# Accurate estimation of cell composition in bulk expression through robust integration of single-cell information

Brandon Jew [1,10], Marcus Alvarez [2,10], Elior Rahmani [3], Zong Miao [1,2], Arthur Ko [2], Kristina M. Garske [2], Jae Hoon Sul [1,4], Kirsi H. Pietiläinen [5,6], Päivi Pajukanta [1,2,7 ✉] & Eran Halperin [2,3,8,9 ✉]

We present Bisque, a tool for estimating cell type proportions in bulk expression. Bisque implements a regression-based approach that utilizes single-cell RNA-seq (scRNA-seq) or single-nucleus RNA-seq (snRNA-seq) data to generate a reference expression profile and learn gene-specific bulk expression transformations to robustly decompose RNA-seq data. These transformations significantly improve decomposition performance compared to existing methods when there is significant technical variation in the generation of the reference profile and observed bulk expression. Importantly, compared to existing methods, our approach is extremely efficient, making it suitable for the analysis of large genomic datasets that are becoming ubiquitous. When applied to subcutaneous adipose and dorso-lateral prefrontal cortex expression datasets with both bulk RNA-seq and snRNA-seq data, Bisque replicates previously reported associations between cell type proportions and measured phenotypes across abundant and rare cell types. We further propose an additional mode of operation that merely requires a set of known marker genes.

[1] Bioinformatics Interdepartmental Program, UCLA, Los Angeles, CA 90095, USA. [2] Department of Human Genetics, David Geffen School of Medicine at UCLA, Los Angeles, CA 90095, USA. [3] Department of Computer Science, School of Engineering, UCLA, Los Angeles, CA 90095, USA. [4] Department of Psychiatry and Biobehavioral Sciences, UCLA, Los Angeles, CA 90095, USA. [5] Obesity Research Unit, Research Program for Clinical and Molecular Metabolism, University of Helsinki, Helsinki 00014, Finland. [6] Obesity Center, Endocrinology Abdominal Center, Helsinki University Central Hospital and University of Helsinki, Helsinki 00260, Finland. [7] Institute for Precision Health, School of Medicine, UCLA, Los Angeles, CA 90095, USA. [8] Department of Anesthesiology, UCLA Health, Los Angeles, CA 90095, USA. [9] Department of Computational Medicine, School of Medicine, UCLA, Los Angeles, CA 90095, USA. [10] These authors contributed equally: Brandon Jew, Marcus Alvarez. ✉ email: ppajukanta@mednet.ucla.edu; ehalperin@cs.ucla.edu

Bulk RNA-seq experiments typically measure total gene expression from heterogeneous tissues, such as tumor and blood samples[1,2]. Variability in cell-type composition can significantly confound analyses of these data, such as in identification of expression quantitative trait loci (eQTLs) or differentially expressed genes[3]. Cell-type heterogeneity may also be of interest in profiling changes in tissue composition associated with disease, such as cancer[4] or diabetes[5]. In addition, measures of cell composition can be leveraged to identify cell-specific eQTLs[6,7] or differential expression[6] from bulk data.

Traditional methods for determining cell-type composition, such as immunohistochemistry or flow cytometry, rely on a limited set of molecular markers and lack in scalability relative to the current rate of data generation[8]. Single-cell technologies provide a high-resolution view into cellular heterogeneity and cell-type-specific expression[9–11]. However, these experiments remain costly and noisy compared to bulk RNA-seq[12]. Collection of bulk expression data remains an attractive approach for identifying population-level associations, such as differential expression regardless of cell-type specificity. Moreover, many bulk RNA-seq studies that have been performed in recent years resulted in a large body of data that is available public databases such as dbGAP and GEO. Given the wide availability of these bulk data, the estimation of cell-type proportions, often termed decomposition, can be used to extract large-scale cell-type-specific information.

There exist a number of methods for decomposing bulk expression, many of which are regression-based and leverage cell-type-specific expression data as a reference profile[13]. CIBERSORT[14] is a SVM-regression-based approach, originally designed for microarray data that utilizes a reference generated from purified cell populations. A major limitation of this approach is the reliance on sorting cells to estimate a reference gene expression panel. BSEQ-sc[15] instead generates a reference profile from single-cell expression data that is used in the CIBERSORT model. MuSiC[16] also leverages single-cell expression as a reference, instead using a weighted non-negative least-squares regression (NNLS) model for decomposition, with improved performance over BSEQ-sc in several datasets.

The distinct nature of the technologies used to generate bulk and single-cell sequencing data may present an issue for decomposition models that assume a direct proportional relationship between the single-cell-based reference and observed bulk mixture. For example, the capture of mRNA and chemistry of library preparation can differ significantly between bulk tissue and single-cell RNA-seq methods, as well as between different single-cell technologies[17,18]. Moreover, some technologies may be measuring different parts of the transcriptome, such as nuclear pre-mRNA in single-nucleus RNA-seq (snRNA-seq) experiments as opposed to cellular and extra-cellular mRNA observed in traditional bulk RNA-seq experiments. As we show later, these differences may introduce gene-specific biases that break down the correlation between cell-type-specific and bulk tissue measurements. Thus, while single-cell RNA-seq technologies have provided unprecedented resolution in identifying expression profiles of cell types in heterogeneous tissues, these profiles generally may not follow the direct proportionality assumptions of regression-based methods, as we demonstrate here.

We present Bisque, a highly efficient tool to measure cellular heterogeneity in bulk expression through robust integration of single-cell information, accounting for biases introduced in the single-cell sequencing protocols. The goal of Bisque is to integrate the different chemistries/technologies of single-cell and bulk tissue RNA-seq to estimate cell-type proportions from tissue-level gene expression measurements across a larger set of samples. Our reference-based model decomposes bulk samples using a single-cell-based reference profile and, while not required, can leverage single-cell and bulk measurements for the same samples for further improved decomposition accuracy. This approach employs gene-specific transformations of bulk expression to account for biases in sequencing technologies as described above. When a reference profile is not available, we propose BisqueMarker, a semi-supervised model that extracts trends in cellular composition from normalized bulk expression samples using only cell-specific marker genes that could be obtained using single-cell data. We demonstrate using simulated and real datasets from brain and adipose tissue that our method is significantly more accurate than existing methods. Furthermore, it is extremely efficient, requiring seconds in cases where other methods require hours; thus, it is scalable to large genomic datasets that are now becoming available.

## Results

**Method overview (Bisque).** A graphical overview of Bisque is presented in Fig. 1. Our reference-based decomposition model requires bulk RNA-seq counts data and a reference dataset with read counts from single-cell RNA-seq. In addition, the single-cell data should be labeled with cell types to be quantified. A reference profile is generated by averaging read count abundances within each cell type in the single-cell data. Given the reference profile and cell proportions observed in the single-cell data, our method learns gene-specific transformations of the bulk data to account for technical biases between the sequencing technologies. Bisque can then estimate cell proportions from the bulk RNA-seq data using the reference and the transformed bulk expression data using non-negative least-squares (NNLS) regression.

**Evaluation of decomposition performance in adipose tissue.** We applied our method to 106 bulk RNA-seq subcutaneous adipose tissue samples collected from both lean and obese individuals, where 6 samples have both bulk RNA-seq and snRNA-seq data available (Table 1). Each of the participants gave a written informed consent. The study protocol was approved by the Ethics Committee at the Helsinki University Hospital, Helsinki, Finland. Adipose tissue consists of several cell types, including adipocytes that are expected to be the most abundant population. Adipose tissue also contains structural cell types (i.e. fibroblasts and endothelial cells) and immune cells (i.e. macrophages and T cells)[19]. These 5 cell-type populations were identified from the snRNA-seq data (Supplementary Fig. 1a).

We observed significant biases between the snRNA-seq and bulk RNA-seq data in samples that had both data available. We found that the linear relationship between the pseudo-bulk (summed snRNA-seq reads across cells) and the true bulk expression varied significantly by each gene (Fig. 2a). Specifically, we observed best fit lines relating these expression levels between technologies with a mean slope of roughly 0.30 and a variance in slope of 5.67. In our model, a slope of 1 would indicate no bias between technologies. We further investigated whether gene expression differences between the bulk and snRNA-seq were the same across individuals and experiments. Comparing log-ratios of RNA-seq to snRNA-seq expression levels, we found that the majority of gene biases were preserved across individuals, tissues, and experiments ($R = 0.75$ across experiments) (Supplementary Fig. 3), providing evidence that technological differences drive consistent gene expression differences across bulk and snRNA-seq methods.

We performed simulations based on the adipose snRNA-seq data to demonstrate the effect of technology-based biases between the reference profile and bulk expression on decomposition performance. In these analyses, we benchmarked Bisque and

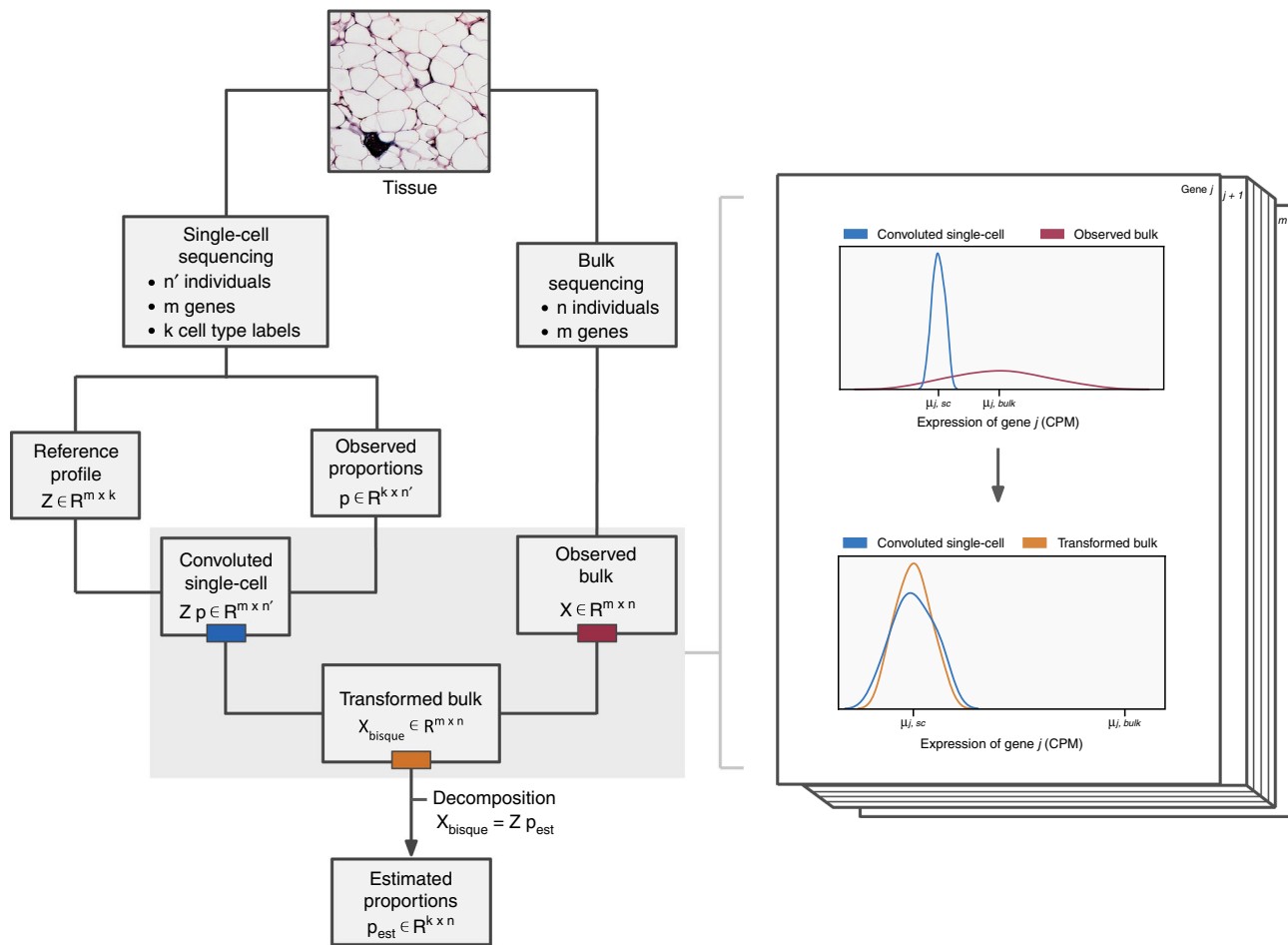

**Fig. 1 Graphical overview of the Bisque decomposition method.** We integrate single-cell and bulk expression by learning gene-specific bulk transformations (pictured on *right*) that align the two datasets for accurate decomposition.

**Table 1 Summary of snRNA-seq and bulk expression datasets used for benchmarking Bisque and existing methods.**

| Tissue | Number of samples | Bulk RNA-seq platform | snRNA-seq platform | snRNA-seq samples | Total nuclei | Average nuclei per individual | Number of cell types |
|---|---|---|---|---|---|---|---|
| Subcutaneous adipose | 106 | Illumina NovaSeq | 10x Genomics Chromium | 6 | 10,947 | 1824 | 5 |
| Dorsolateral prefrontal cortex | 636 | Illumina HiSeq | 10x Genomics Chromium | 8 | 68,028 | 8503 | 11 |

three existing methods (MuSiC, BSEQ-sc, and CIBERSORT). Briefly, we simulated bulk expression for 6 individuals by summing the observed snRNA-seq read counts. To model discordance between the reference and bulk, we applied gene-specific linear transformations of the simulated bulk expression. For each gene, the coefficient and intercept of the linear transformation were sampled from half-normal distributions with increasing variance. In this model, a higher variance corresponds to a larger bias between sequencing experiments. Although these transformations closely mirrored the Bisque decomposition model, they utilized the true snRNA-seq counts for each individual whereas Bisque learned these transformations using the reference profile generated from averaging these counts across all cells. Hence, this simulation framework introduced additional noise that Bisque does not entirely model. We evaluated decomposition performance by comparing proportion estimates to the proportions observed in the snRNA-seq data in terms of global Pearson correlation ($R$) and root-mean squared

deviation (RMSD). Owing to the small number of samples, we applied leave-one-out cross-validation to predict the cell composition of each individual using the remaining snRNA-seq samples as training data for each method. In these simulations, Bisque remained robust ($R \approx 0.85$, RMSD $\approx 0.07$) at higher levels of simulated bias between the bulk and snRNA-seq-based reference (Fig. 2b).

Next, we performed this cross-validation benchmark on the observed bulk RNA-seq data for these 6 individuals and found that Bisque ($R = 0.923$, RMSD $= 0.074$) provided significantly improved global accuracy in detecting each cell type over existing methods (Table 2, Supplementary Fig. 1b). MuSiC ($R = -0.111$, RMSD $= 0.427$), BSEQ-sc ($R = -0.113$, RMSD $= 0.432$), and CIBERSORT ($R = -0.131$, RMSD $= 0.416$) severely underestimated the proportion of adipocytes (the most abundant population in adipose tissue) while overestimating the endothelial cell fraction. We also benchmarked CIBERSORTx[20], which employs a batch correction mode to account for biases in

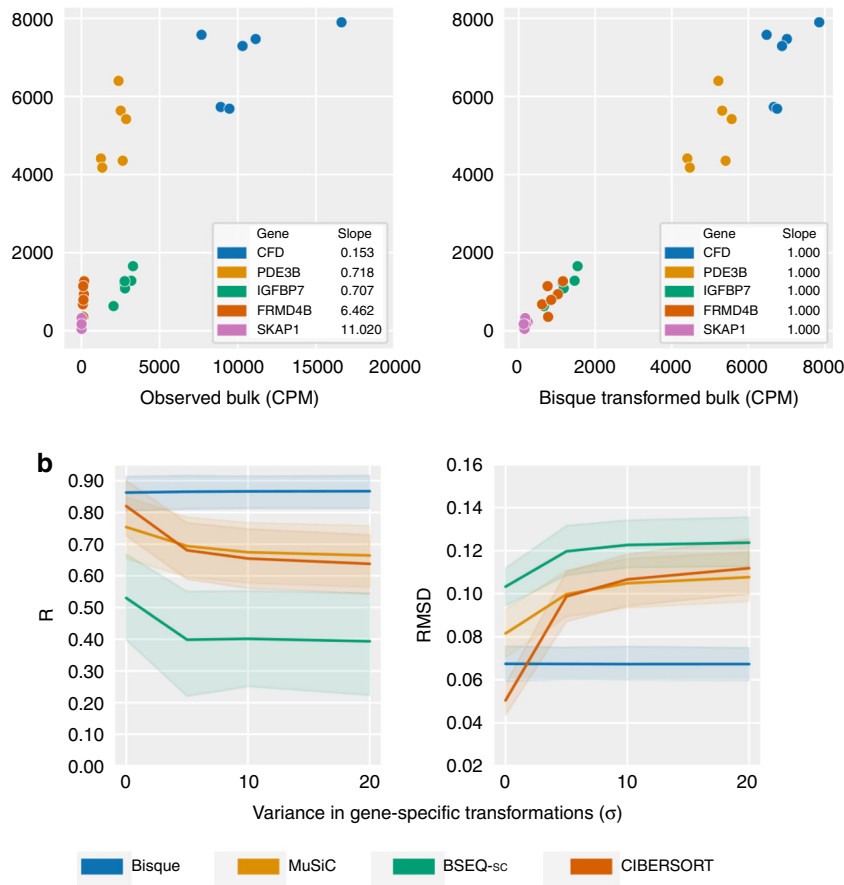

**Fig. 2 The effect of discrepancies between a single-cell-based reference and bulk expression on decomposition. a** Observed discrepancies in real data between single-nucleus and bulk expression for selected marker genes (*left*) for six individuals. Each color corresponds to a gene. On the left, observed bulk expression on the *x* axis is plotted against the pseudo-bulk expression on the *y* axis, where pseudo-bulk expression is calculated by summing the single-cell-based reference with cell proportions as weights. On the right, the Bisque transformation of bulk expression is on the *x* axis. Bisque recovers a one-to-one relationship by transforming the bulk expression for improved decomposition accuracy (*right*). **b** Simulation of bulk expression for six individuals based on true adipose snRNA-seq data with increasing gene-specific differences. These differences are modeled as a linear transformation of the summed snRNA-seq counts with coefficient and intercept sampled from half-normal distributions with parameter as indicated on the *x* axis. At $\sigma = 0$, the simulated bulk is simply the sum of the observed single-cell read counts. Performance on *y* axis measured in global Pearson correlation (*R*) (*left*) and root-mean squared deviation (RMSD) (*right*). *Shaded regions* indicate 95% confidence intervals based on bootstrapping with *central lines* indicating the mean observed value. Bisque remains robust to increasing gene-specific variation between single-cell and bulk expression levels. Source data are provided as a Source Data file.

**Table 2 Leave-one-out cross-validation in subcutaneous adipose using 6 samples with snRNA-seq and bulk RNA-seq data available.**

| Method | R | RMSD |
|---|---|---|
| Bisque | **0.923 ± 0.064** | **0.074 ± 0.034** |
| CIBERSORTx | 0.687 ± 0.450 | 0.099 ± 0.046 |
| MuSiC | −0.111 ± 0.182 | 0.427 ± 0.058 |
| BSEQ-sc | −0.113 ± 0.180 | 0.432 ± 0.058 |
| CIBERSORT | −0.131 ± 0.176 | 0.416 ± 0.059 |

Proportions based on snRNA-seq were used as a proxy for the true proportions. Performance measured in Pearson correlation (*R*) and root-mean-square deviation (RMSD) across all 5 identified cell types in each sample. Reported values were averaged across the 6 samples with standard deviation indicated. Bold values indicate the highest performing method with respect to each metric. Source data are provided as a Source Data file.

sequencing technologies. Although CIBERSORTx ($R = 0.687$, RMSD = 0.099) outperformed existing methods, Bisque provided improved accuracy. It should be noted that cell-specific accuracy is more informative than global *R* and RMSD; however, these small sample sizes did not provide robust measures of within-cell-

type performance in this cross-validation framework (Supplementary Fig. 1c). We were able to slightly improve the number of detected cell populations by MuSiC, BSEQ-sc, and CIBERSORT when we considered only snRNA-seq reads aligning to exonic regions of the transcriptome, indicating that intronic reads introduced increasing discrepancy between snRNA-seq and bulk RNA-seq in the context of decomposition. However, given that a significant portion of the nuclear transcriptome consists of pre-mRNA, this filtering process removed over 40% of cells detected in the snRNA-seq data. Moreover, Bisque provided improved accuracy over existing methods using this exonic subset of the snRNA-seq data (Supplementary Fig. 1d).

We then applied these decomposition methods to the remaining 100 bulk samples and found that the distribution of cell-proportion estimates produced by Bisque were most concordant with the expected distribution inferred from the limited number of snRNA-seq samples and previously reported proportions[21,22] (Fig. 3a). Although these benchmarks provided a measure of calibration (i.e. the ability to detect cell populations in expected ranges), they did not provide measurements of cell-specific proportion accuracy across individuals. In order to

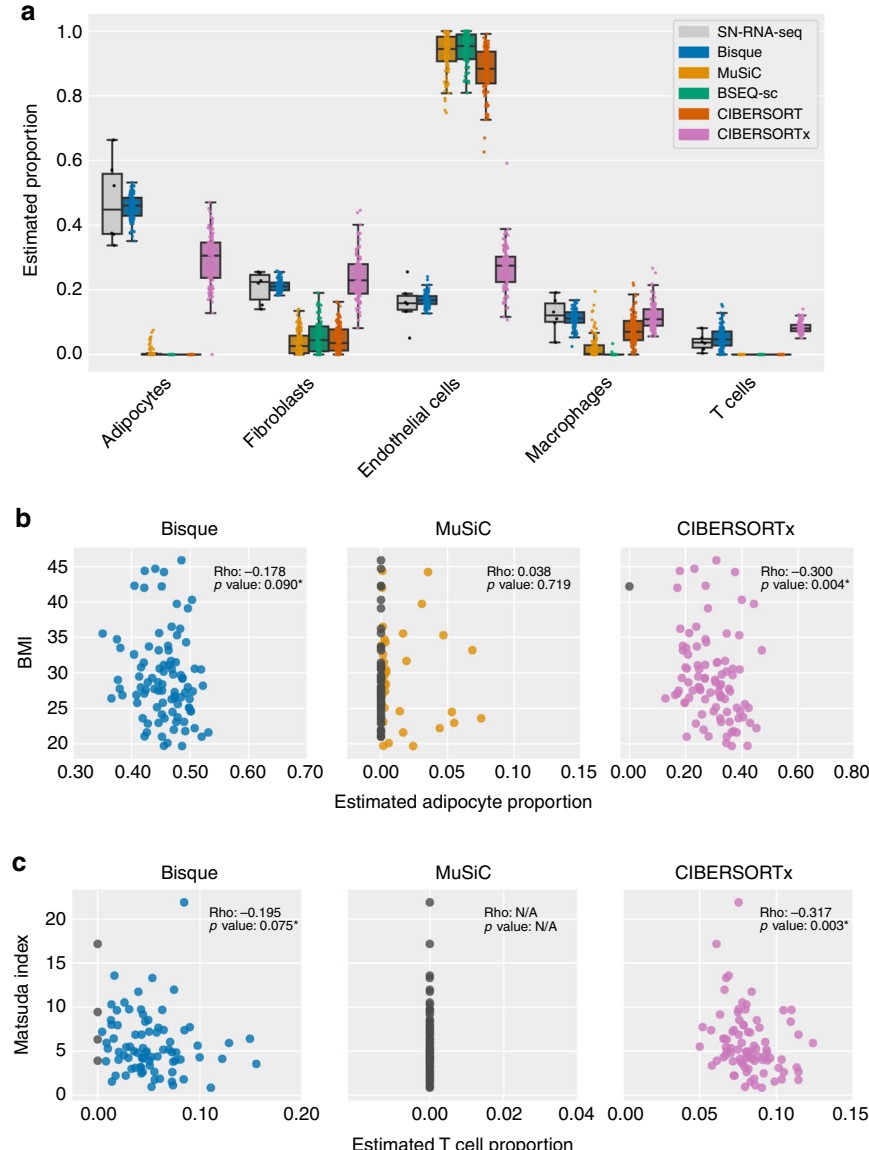

**Fig. 3 Decomposition benchmark in human subcutaneous adipose tissue. a** Comparison of decomposition estimates from 100 individuals with estimates from 6 individuals with snRNA-seq data available. Each color represents a benchmarked method. *Boxes* indicate the quartiles of the estimated proportions with *whiskers* extending 1.5 times the interquartile range. *Points* are individual samples that are represented by the boxplot. **b**, **c** Scatterplots comparing decomposition estimates with measured phenotypes in 100 individuals. Reported 'rho' corresponds to Spearman correlation and *p*-values indicate the significance of these correlations, with an *asterisk* denoting significance after correction for covariates in a linear-mixed model. CIBERSORT and BSEQ-sc are not shown as they did not detect these cell populations. These examples include the most abundant (adipocytes) and rarest (T cells) cell types identified in the snRNA-seq data. Significance of associations reported in Supplementary Table 1. **b** Adipocyte proportion has been observed to negatively correlate with BMI so we expected a negative correlation. Bisque ($p = 0.030$) and CIBERSORTx ($p = 0.001$) produced significant negative associations after correcting for sex, age, age-squared, and relatedness in a linear-mixed model. **c** T-cell proportion has previously been reported to positively correlate with insulin resistance. Matsuda index decreases with higher insulin resistance so we expected a negative correlation. Bisque ($p = 0.002$) and CIBERSORTx ($p = 0.046$) produced significant negative associations after correcting for diabetes status, sex, age, age-squared, and relatedness in a linear-mixed model. Source data are provided as a Source Data file.

evaluate cell-specific accuracy, we replicated previously reported associations between cell proportions and measured phenotypes. Specifically, we compared cell-proportion estimates from each method to body mass index (BMI) and Matsuda index, a measure of insulin resistance. We measured the significance of these association based on *t*-values estimated in a linear-mixed model accounting for age, age-squared, and sex as fixed effects and relatedness as a random effect.

Obesity is associated with adipocyte hypertrophy, the expansion of the volume of fat cells[23]; thus, we expected a negative

association between adipocyte proportion and BMI. Bisque, MuSiC, and CIBERSORTx produced adipocyte proportion estimates that replicate this behavior, while BSEQ-sc and CIBERSORT were unable to detect this cell population (Fig. 3b). The adipocyte proportion estimates produced by Bisque ($p = 0.030$) and CIBERSORTx ($p = 0.001$) had a significant negative association with BMI (Supplementary Table 1a). In addition, macrophage abundance has been shown to increase in adipose tissue with higher levels of obesity, concomitant with a state of low grade inflammation[24]. Each method detected macrophage

populations that positively associated with BMI; however, only Bisque ($p < 0.001$), BSEQ-sc ($p = 0.004$), and CIBERSORTx ($p = 0.034$) reached significance (Supplementary Table 1b).

T cells were the least abundant cell-type population identified from the snRNA-seq data, constituting around 4% of all sequenced nuclei. The abundance of T cells has been observed to positively correlate with insulin resistance[25]. Thus, we compared decomposition estimates for T-cell proportions to Matsuda index. As a lower Matsuda index indicates higher insulin resistance, we expect a negative association between T-cell proportion and Matsuda index. Proportion estimates produced by Bisque and CIBERSORTx followed this trend while the remaining existing methods did not identify T cells in the bulk samples (Fig. 3c). We found this association significant for Bisque ($p = 0.002$) and CIBERSORTx ($p = 0.046$) (Supplementary Table 1c) after correcting for diabetes status, as Matsuda index may not be informative in these individuals[26].

**Evaluation of decomposition performance in cortex tissue.** We also benchmarked these decomposition methods using expression data collected from the dorsolateral prefrontal cortex (DLPFC). This dataset was generated by the Rush Alzheimer's Disease (AD) Center[27] and includes 636 postmortem bulk RNA-seq samples. The Religious Orders Study and Rush Memory and Aging Project were approved by an IRB of Rush University Medical Center. Both bulk RNA-seq and snRNA-seq data were collected from 8 of the individuals (Table 1). Using the same pipeline we used to process the adipose dataset, we identified 11 clusters: 3 neuronal subtypes, 2 interneuronal subtypes, 2 astrocyte subtypes, oligodendrocytes, oligodendrocyte progenitor cells, and microglia (Supplementary Fig. 2a). We observed a higher overlap in marker genes for these clusters than in those identified in the adipose dataset (average of 10% of marker genes shared between clusters in DLPFC compared to 3% in adipose) (Supplementary Fig. 4a, b).

We again applied leave-one-out cross-validation on the 8 individuals with both RNA-seq and snRNA-seq data available. In this example, we randomly sampled 25% of the nuclei in the snRNA-seq data to accommodate CIBERSORTx (which is currently web-based and restricts the size of files that can be processed). Bisque was able to detect each cell population identified from the snRNA-seq data with high global accuracy ($R = 0.924$, RMSD = 0.029) while MuSiC ($R = −0.192$, RMSD = 0.173), BSEQ-sc ($R = 0.098$, RMSD = 0.120), and CIBERSORT ($R = −0.281$, RMSD = 0.197) did not detect a number of cell populations (Table 3, Supplementary Fig. 2b, c). Bisque also provided higher accuracy than CIBERSORTx ($R = 0.671$, RMSD = 0.070). However, we found that the performance of the

existing methods improved when estimates with subtypes were summed together (Supplementary Fig. 2d). Although each method was able to quantify major cell populations after merging subtypes, Bisque was able to distinguish between these closely related cell populations. Interestingly, we found that in both adipose and DLPFC, endothelial cell proportions were overestimated by each of the existing methods.

We applied these decomposition methods to the remaining 628 individuals and compared the distribution of estimates to the proportions observed in the 8 snRNA-seq samples. We found that Bisque was able to detect each cell population and produced estimates that were closest in mean to the snRNA-seq observations (Fig. 4a). The increased accuracy of Bisque over existing methods persisted when we merged closely related subtypes (Supplementary Fig. 2e). Moreover, immunohistochemistry (IHC) analyses on a 70 of these samples found similar proportions of major cell populations[28], confirming the relative accuracy of snRNA-seq-based estimates of cell proportions.

Again, to determine cell-specific decomposition accuracy, we replicated known associations between cell-type proportions and measured phenotypes in the 628 individuals. For these analyses, we compared cell-proportion estimates to each individual's Braak stage and physician cognitive diagnostic category at time of death. Braak stage is a semiquantitative measure of neurofibrillary tangles, ranging in value from 0 to 5 with increasing severity. The cognitive diagnostic category provides a semiquantitative measure of dementia severity, where a code of 1 indicates no cognitive impairment and 5 indicates a confident diagnosis of AD by physicians. We determined the significance of these associations based on $t$-values estimated by a linear regression model that accounted for age, age-squared, and sex.

Neuronal death is a hallmark symptom of AD[29]. Therefore, we expected to find a negative association between cognitive diagnosis and neuron proportion. We found that each decomposition method provides estimates of total neuron proportion that tend to decrease with cognitive diagnostic category (Fig. 4b). Each method generates proportions with negative association with cognitive diagnosis. Each method, excluding BSEQ-sc, reached significance in this model ($p \leq 0.001$ for each method) (Supplementary Table 2a). As another example, we compared each individual's Braak stage to their estimated proportion of microglia, a relatively small cell population that constituted roughly 5% of the sequenced nuclei. Microglia activation has been observed to increase with AD severity[30]. We used Braak stage as a proxy for AD severity and expected a positive association between microglia proportion and Braak stage. Bisque and MuSiC provided estimates that follow this expected trend (Fig. 4c). Only Bisque produced estimates with a significant positive association ($p = 0.001$) (Supplementary Table 2b). Interestingly, we observed a decrease in microglia proportions estimated by Bisque in Braak stage 6 individuals, which has been previously observed in AD patients[31].

**Table 3 Leave-one-out cross-validation in dorsolateral prefrontal cortex using 8 samples with snRNA-seq and bulk RNA-seq data available.**

| Method | R | RMSD |
|---|---|---|
| Bisque | **0.924 ± 0.062** | **0.029 ± 0.010** |
| CIBERSORTx | 0.671 ± 0.153 | 0.070 ± 0.019 |
| MuSiC | −0.192 ± 0.107 | 0.173 ± 0.013 |
| BSEQ-sc | 0.098 ± 0.216 | 0.120 ± 0.023 |
| CIBERSORT | −0.281 ± 0.049 | 0.197 ± 0.012 |

Proportions based on snRNA-seq were used as a proxy for the true proportions. Performance measured in Pearson correlation ($R$) and root-mean-square deviation across all 11 identified cell types in each sample. Reported values were averaged across the 8 samples with standard deviation indicated. We performed these experiments with 25% of the snRNA-seq data in order to accommodate the file size limit of the current web-based implementation of CIBERSORTx. Bold values indicate the highest performing method with respect to each metric. Source data are provided as a Source Data file.

**Runtime comparison of reference-based decomposition methods.** Given the large amounts of transcriptomic data that are becoming available, we also benchmarked these decomposition methods in terms of runtime. In the subcutaneous adipose dataset, which included 100 bulk RNA-seq samples and 6 snRNA-seq samples with about 1800 nuclei sequenced per individual, Bisque was able to estimate cell proportions efficiently compared to existing methods. Bisque (1 s) and MuSiC (1 s) provided decomposition estimates faster than BSEQ-sc (26 s), CIBERSORT (27 s), and CIBERSORTx (389 s) (Fig. 5a). Bisque also provided improved efficiency in processing the reduced DLPFC dataset, which included 628 bulk RNA-seq samples and 8

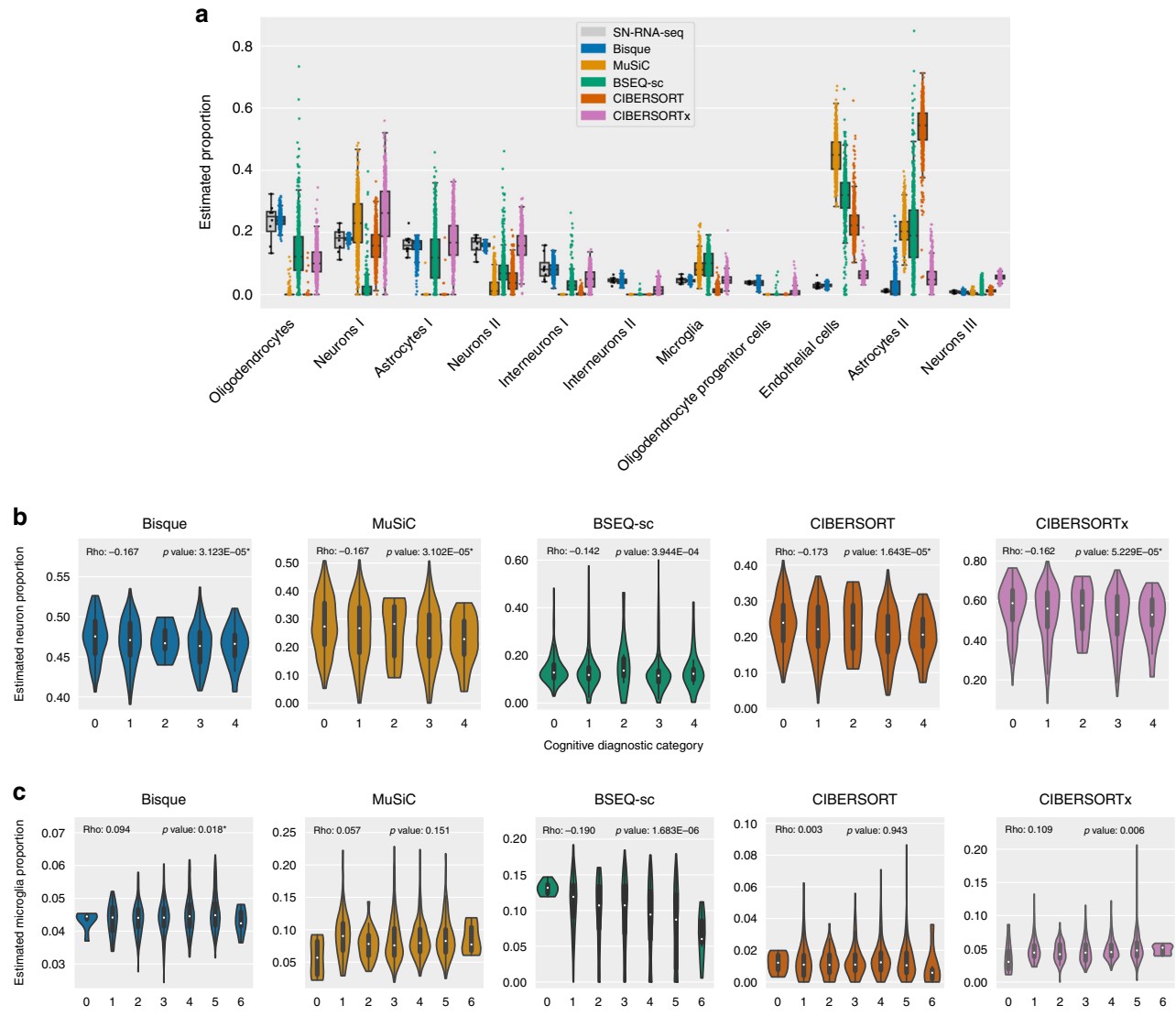

**Fig. 4 Decomposition benchmark in human dorsolateral prefrontal cortex tissue.** We randomly sampled 25% of the nuclei in the snRNA-seq data to accommodate the file size limit of the web-based implementation of CIBERSORTx at the time of writing. **a** Comparison of decomposition estimates from 628 individuals with estimates from 8 individuals with snRNA-seq data available. Each color represents a benchmarked method. *Boxes* indicate the quartiles of the estimated proportions with *whiskers* extending 1.5 times the interquartile range. *Points* are individual samples that are represented by the boxplot. **b, c** Violin plots depicting association of decomposition estimates aggregated into major cell types with measured phenotypes in 628 individuals. Reported 'rho' corresponds to Spearman correlation and *p*-values indicate the significance of these correlations, with an *asterisk* denoting both an expected effect direction and significance after correction for covariates. Examples shown are for the most abundant (neurons) and least abundant (microglia) populations detected in the snRNA-seq data. Significance of associations reported in Supplementary Table 2. **b** Neuronal degeneration has been observed in patients diagnosed with Alzheimer's disease (AD). Cognitive diagnostic category measures a physician's diagnosis of cognitive impairment (CI), with 0 indicating no CI and 4 indicating a confident AD diagnosis. We expected a negative correlation between neuron proportion and cognitive diagnostic category. **c** Microglia proportion has been observed to positively correlate with increased severity of AD symptoms, such as neurofibrillary tangles. Braak stage provides a semiquantitative measure of tangle severity, so we expected an overall positive correlation between microglia proportion and Braak stage. In addition, a decrease in microglia abundance has been previously reported at Braak stages 5 through 6 in AD patients. Only Bisque produced estimates with a significant positive association ($p = 0.001$) after correcting for sex, age, and age-squared in a linear regression model. Source data are provided as a Source Data file.

snRNA-seq samples with around 2125 nuclei per individual. Bisque (4 s) and MuSiC (10 s) estimated cell proportions relatively quickly compared to BSEQ-sc (273 s), CIBERSORT (298 s), and CIBERSORTx (6566 s) (Fig. 5b).

**Robustness of the reference-based decomposition model.** Our reference-based decomposition method is based on the assumption that cell populations are equally represented in single-cell and bulk

RNA sequencing of the same tissue samples. As this assumption may be violated[32], we explored the performance of our model as we relaxed this assumption in simulations. First, we simulated snRNA-seq data where cell proportions were increasingly biased. Using the DLPFC snRNA-seq data, we downsampled or upsampled the cells identified as microglia at varying levels and performed decomposition. Indeed, the absolute estimates produced by Bisque propagated these shifts in snRNA-seq proportions. However, we found that our estimated microglia proportions, regardless of these shifts,

**a**

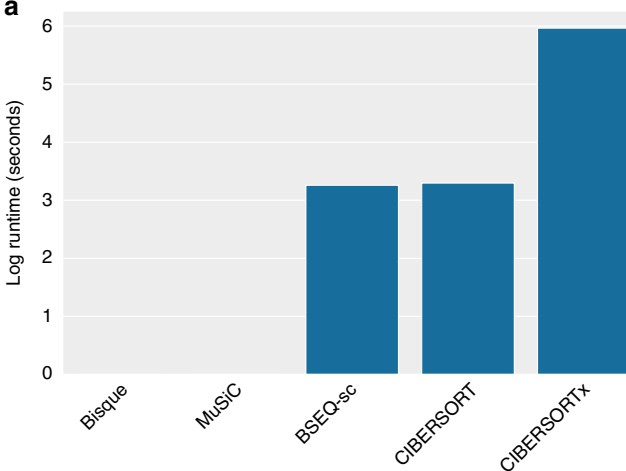

**b**

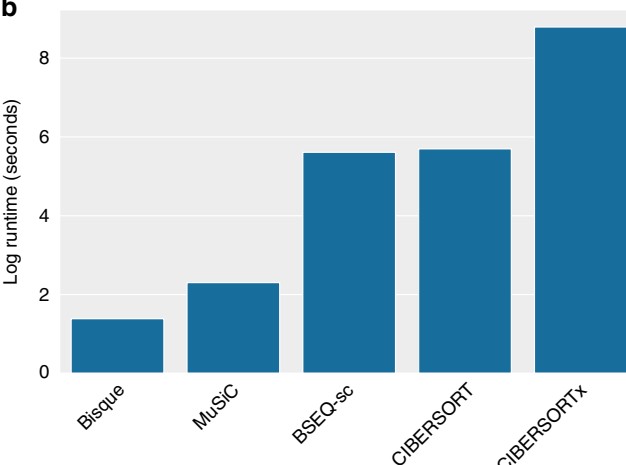

**Fig. 5 Runtime comparisons in log-transformed seconds for benchmarked reference-based decomposition methods. a** Runtime for subcutaneous adipose dataset, which included 100 RNA-seq samples and 6 snRNA-seq samples with around 1800 nuclei per individual. **b** Runtime for dorsolateral prefrontal cortex dataset, which included 628 RNA-seq samples and 8 snRNA-seq samples. We benchmarked each method using around 2125 nuclei per snRNA-seq sample. Source data are provided as a Source Data file.

maintained an expected positive association with Braak stage. This positive association served as evidence for the correlation between these estimates and the true microglia proportions (Supplementary Fig. 5a). Given these results, we suggest that users take note of this behavior if both the mean abundances are important for downstream analysis and the single-cell reference data is known to be significantly biased against specific cell populations of interest.

Next, we simulated a situation where an unknown cell population contributes to bulk expression but is not represented in the snRNA-seq reference data. For situations where this unknown contribution varies across the bulk dataset, we simulated bulk expression by mixing the observed bulk expression for the DLPFC dataset with increasing amounts of expression observed in the adipose dataset. To determine the effect of unknown cell populations on our model, we analyzed the distribution of residual norms produced by the method. These residual norms provide a measure of the difference between the vector of observed bulk and expression reference weighted by the estimated proportions across all genes for each individual. As we increased the contribution from unknown cell types, the

residual norm values tend to increase (Supplementary Fig. 5b). In our simulation framework, this variability in unknown cell-type contribution could be qualitatively identified by the presence of a multimodal residual norm distribution.

Given that single-cell datasets still remain relatively small compared to bulk datasets, we also explored the impact of sample size in the reference single-cell data on the performance of Bisque. In the DLPFC dataset, we saw a drop in performance when using less than four randomly selected snRNA-seq samples (Supplementary Fig. 5c). This threshold is likely to differ between experiments, though we recommend at least three single-cell samples to generate reference data.

Finally, as marker gene selection can vary between studies, we were interested in the performance of Bisque as we varied the number of marker genes. Again, we measured cell-type proportion estimation performance for microglia in the DLPFC dataset by correlating the estimates with Braak stage, which is known to have a positive association. We recalculated this correlation as we removed marker genes for this cell type. We removed marker genes in order of both decreasing and increasing log-fold change, which provides a measure of the importance of marker genes for identifying this cell type. In both procedures, we observe that as we remove an increasing percentage of the 102 identified marker genes, performance remains stable until a shared drop off point around 75% (Supplementary Fig. 5d). As we observed this trend in both marker gene removal schemes, we assume that a relatively few number of marker genes, regardless of their log-fold change magnitude, can be used to accurately estimate cell-type proportions. These results suggest that as long as a core set of marker genes are present, variations in less important marker genes will have little effect on downstream analyses.

**Marker-based decomposition using cell-type marker genes.** Although a reference profile from snRNA-seq can help to decompose bulk-level gene expression, it may not be available for the same dataset. The majority of bulk RNA-seq datasets do not have corresponding snRNA-seq data in the same set of individuals. However, marker gene information from prior experiments can still be applied to distinct expression datasets of the same tissue. The basis of most decomposition methods relies on the logic that as the proportion of a cell type varies across individuals, the expression of its marker genes will tend to correlate in the same direction as its cell-type proportion. This linear co-variation can be captured in a principal components analysis (PCA). Under the same argument, the more cell-type-specific a marker gene is, the more its expression will reflect its cell-type proportion. These observations form the basis for BisqueMarker, a weighted PCA-based (wPCA) decomposition approach. Genes that are more specifically expressed within a cell type will provide more information than genes with shared expression across cell types. To estimate cell-type proportions without the use of cell-type-specific gene expression information, we applied wPCA to bulk-level adipose tissue expression.

For each cell type, we extracted the first PC from a wPCA of the expression matrix of its markers. The expression matrix was corrected for the first global expression PC as a covariate so that wPCA estimates would not reflect technical variation. We first confirmed that these genes were distinct across cell types. If 2 cell types share a high proportion of marker genes, the wPCA estimates from bulk RNA-seq will correlate highly. We then investigated whether the second or third PC could have represented cell-type proportions. The percent of variance explained by the first PC was typically 30–60% across adipose cell types, and additionally, over 90% of the markers correlated in the same direction as the first PC. In contrast, roughly 50–70% of

markers correlated in the same direction as the second or third PC. As performed for reference-based decomposition, we correlated phenotypes with cell-type proportions estimated by BisqueMarker. We identified the same associations as with reference-based decomposition, demonstrating its validity when a reference is not available (Supplementary Table 1). Similarly, we observed the same trends between estimated cell-type abundances and phenotypes as we did using our reference-based method in the DLPFC cohort (Supplementary Table 2).

## Discussion

Bisque effectively leverages single-cell information to decompose bulk expression samples, outperforming existing methods in datasets with snRNA-seq data available. In simulations, we demonstrated that the decomposition accuracy of Bisque is robust to increasing variation between the generation of the reference profile and bulk expression, which is a significant issue when comparing snRNA-seq and bulk RNA-seq data. In observed bulk expression, our reference-based method accurately estimates cell proportions that are consistent with previously reported distributions and reliably detects rare cell types. We found that these estimates consistently follow expected trends with measured phenotypes, suggesting that cell-specific estimates of proportion are sufficiently accurate to extract relevant biological signals. In addition, differences in tissue structure can lead to significant differences in the quality of single-cell expression data[33]. We demonstrated the improved performance of our method in adipose and DLPFC, two distinct tissues, suggesting that Bisque is robust across different tissue types.

The cell-type proportion estimates determined by Bisque may be utilized to effectively identify cell-type-specific interactions, such as expression quantitative trait loci (eQTLs), and adjust for confounding effects from variability in cell populations. With this reference-based approach, single-cell sequencing of a subset of samples from large-scale bulk expression cohorts can provide high power to detect cell-specific associations in complex phenotypes and diseases.

However, we note that there are limitations to this reference-based method that users should consider. First, if the number of individuals with single-cell data available is small, the reference profile and gene-specific transformations may become unreliable. In addition, a key assumption of our transformation framework is that single-cell-based estimates of cell proportions accurately reflect the true proportions we wish to estimate. As a result of this assumption, Bisque provides estimates of cell proportions reported by the single-cell technology used to generate the reference data. Given that snRNA-seq can provide less bias in isolating specific cell types compared to scRNA-seq[34,35], we expect these estimates to be useful for downstream analyses such as those previously discussed. Nevertheless, the accuracy of Bisque may decrease if the proportion of cell types captured by single-cell experiments differs significantly from the true physiological distributions. Therefore, we advise users to take caution if there is a known significant bias in the single-cell measurements of a tissue, such as severe underrepresentation of a cell type of interest[32,35], that can affect downstream analysis. Our results demonstrate that even with these limitations, Bisque can be used to provide cell-type specific biological insight in relevant datasets.

In cases where these described issues may be significant, BisqueMarker provides cell-type abundance estimations using only known marker genes. Although this reference-free method may be less accurate than reference-based methods, it does not depend on single-cell based estimates of cell proportions or expression profiles, but rather on the fact that the expression in certain genes differs across different cell types; moreover, this method also does not model explicitly the expression level, and it is thus robust to biases in the single-cell sequencing protocol. We found that BisqueMarker estimates followed expected trends with measured phenotypes; however, it should be noted that this method estimates relative differences in abundances that cannot be compared across cell types. Also, given the semi-supervised nature of this method, these cell-type abundance estimates may include signals from technical or other biological variation in the data. Therefore, we highly suggest applying this method to data that is properly normalized with sources of undesired variation removed.

## Methods

**Processing bulk expression data.** Paired-end reads were aligned with STAR v2.5.1 using default options. Gene counts were quantified using featureCounts v1.6.3. For featureCounts, fragments were counted at the gene-name level. Alignment and gene counts were generated against the GRCh38.p12 genome assembly. STAR v2.5.1 and GRCh38.p12 were included with CellRanger 3.0.2, which was used to process the single-nucleus data.

**Processing single-nucleus expression data.** Reads from single nuclei sequenced on the 10x Genomics Chromium platform were aligned and quantified using the CellRanger 3.0.2 count function against the GRCh38.p12 genome assembly. To account for reads aligning to both exonic and intronic regions, each gene transcript in this reference assembly was relabeled as an exon as CellRanger counts exonic reads only. We perform this additional step since snRNA-seq captures both mature mRNA and pre-mRNA, the latter of which includes intronic regions.

After aggregating each single-nucleus sample with the CellRanger aggr function, the full dataset was processed using Seurat v3.0.0[36]. The data were initially filtered for genes expressed in at least 3 cells and filtered for cells with reads quantified for between 200 and 2500 genes. We further filtered for cells that had percentage of counts coming from mitochondrial genes less than or equal to 5%. The data were normalized, scaled, and corrected for mitochondrial read percentages with sctransform v0.2.0[37] using default options.

To identify clusters, Seurat employs a shared nearest neighbor approach. We identified clusters using the top 10 principal components of the processed expression data with resolution set at 0.2. The resolution parameter controls the number of clusters that will be identified, and suggested values vary depending on the size and quality of the dataset. We chose a value that produced 6 clusters in the adipose dataset and 13 clusters in the DLPFC dataset and visualized the clustering results with UMAP[38].

Marker genes were identified by determining the average log-fold change of expression of each cluster compared to the rest of the cells. We identified marker genes as those with an average log-fold change above 0.25. The significance of the differential expression of these genes was determined using a Wilcoxon rank sum test. Only genes that were detected in at least 25% of cells were considered. Clusters with many mitochondrial genes as markers (nine genes detected in both datasets) were removed from both datasets. In addition, a cluster with only three marker genes was removed from the DLPFC datasets. Finally, we remove mitochondrial genes from the list of marker genes for decomposition as we assume reads aligning to the mitochondrial genome originate from extra-nuclear RNA in the snRNA-seq dataset (targeting nuclear RNA).

Clusters were labeled by considering cell types associated with the identified marker genes. Marker genes were downloaded from PanglaoDB[39] and filtered for entries validated in human cells. For each gene, we count the possible cell-type labels. Each cluster was labeled as the most frequent cell type across all of its marker genes, with each label associated with a gene weighted by the average log-fold change. If multiple clusters shared a cell-type label, we consider each cluster a subtype of this label.

Exon-aligned reads were processed in the same exact procedure but snRNA-seq data was aligned to just exonic regions. Cluster names were manually changed for both datasets when aligned to exons to match the clusters from intronic reads as well. Specifically, for clusters identified in the exonic data not found in the full data, we relabeled as the label with the highest score found in the full data. These relabeled clusters were similar in proportion to the corresponding cluster in the full dataset.

**Bisque reference-based decomposition model.** We assume that only a subset of genes are relevant for estimating cell-type composition. For the adipose and DLPFC datasets, we selected the marker genes identified by Seurat as described previously. Moreover, we filter out genes with zero variance in the single-cell data, unexpressed genes in the bulk expression, and mitochondrial genes. We convert the remaining gene counts to counts-per-million to account for variable sequencing depth. For $m$ genes and $k$ cell types, a reference profile $\mathbf{Z} \in \mathbf{R}^{m \times k}$ is generated by averaging relative abundances within each cell type across the entire single-cell dataset.

Although there is a strong positive correlation between bulk and single-cell-based pseudo-bulk (summed single-cell counts) expression data, we observe that

the relationship is not one-to-one and varies between genes. This behavior indicates that the distribution of observed bulk expression may significantly differ from the distribution of the single-cell profile weighted by cell proportions. We propose transforming the bulk data to maximize the global linear relationship across all genes for improved decomposition. Our goal is to recover a one-to-one relationship between the transformed bulk and expected convolutions of the reference profile based on single-cell based estimates of cell proportions. This transformed bulk expression better satisfies the assumptions of regression-based approaches under sum-to-one constraints.

Cell-type proportions $\mathbf{p} \in \mathbf{R}^{k \times n'}$ are determined by counting the cells with each label in the single-cell data for $n'$ individuals. Given these proportions and the reference profile $\mathbf{Z}$, we calculate the pseudo-bulk for the single-cell samples as $\mathbf{Y} = \mathbf{Z}\mathbf{p}$, where $\mathbf{Y} \in \mathbf{R}^{m \times n'}$. For each gene $j$, our goal is to transform the observed bulk expression across all $n$ bulk samples $\mathbf{X}_j \in \mathbf{R}^n$ to match the mean and variance of $\mathbf{Y}_j \in \mathbf{R}^{n'}$; hence, the transformation of $\mathbf{X}_j$ will be a linear transformation.

If individuals with both single-cell and bulk expression are available, we fit a linear regression model to learn this transformation. Let $\mathbf{X}'_j \in \mathbf{R}^{n'}$ denote the expression values for these $n'$ overlapping individuals. We fit the following model (with an intercept) and apply the model to the remaining bulk samples as our transformation:

$$Y_j = \beta_j X'_j + \epsilon_j \tag{1}$$

If there are no single-cell samples that have bulk expression available, we assume that the observed mean of $\mathbf{Y}_j$ is the true mean of our goal distribution for the transformed $\mathbf{X}_j$. We further assume that the sample variance observed in $\mathbf{Y}_j$ is larger than the true variance of the goal distribution, as the number of single-cell samples is typically small. We use a shrinkage estimator of the sample variance of $\mathbf{Y}_j$ that minimizes the mean squared error and results in a smaller variance than the unbiased estimator:

$$\hat{\sigma}_j^2 = \frac{1}{n'+1} \sum_{i=1}^{n'} (Y_{i,j} - \bar{Y}_j)^2 \tag{2}$$

We transform the remaining bulk as follows:

$$X_{j,\text{transformed}} = \frac{X_j - \bar{X}_j}{\sigma_{X_j}} \hat{\sigma}_j + \bar{Y}_j \tag{3}$$

where a bar indicates the mean value of the observed data and $\sigma_{\mathbf{X}_j}$ is the unbiased sample variance of $\mathbf{X}_j$.

To estimate cell-type proportions, we apply non-negative least-squares regression with an additional sum-to-one constraint to the transformed bulk data. For individual $i$, we minimize the following with respect to the cell-proportion estimate $\mathbf{p}_i$:

$$||Zp_i - X_{i,\text{transformed}}||_2 \, s.t. \, p_i \geq 0, \, \sum p_i = 1 \tag{4}$$

**Simulating bulk expression based on single-nucleus counts**. We simulate the base bulk expression as the sum of all counts across cells/nuclei sequenced from an individual. To introduce gene-specific variation between the bulk and single-cell data, we sample a coefficient $\beta_j$ and an intercept $\alpha_j$ from a half-normal (HN) distributions:

$$\beta_j \sim \text{HN}(\sigma) + 1 \tag{5}$$

$$\alpha_J \sim \text{HN}(\sigma) \tag{6}$$

At $\sigma = 0$, the base simulated bulk expression remains unchanged. We used a HN distribution to ensure coefficients and intercepts are positive. Although our method can handle negative coefficients, this simulation model assumes expression levels have a positive correlation across technologies. We performed 10 replicates of this data-generating process at each $\sigma$ in 0, 5, 10, 20. Decomposition performance on these data were measured in terms of global R and RMSD and plotted with 95% confidence intervals based on bootstrapping.

**Measuring significance of cell proportion-trait association**. Reported associations were measured in terms of Spearman correlation. To determine the statistical significance of these associations while accounting for possible confounding factors, we applied two approaches. For the adipose dataset, which consisted entirely of twin pairs, we applied a linear-mixed-effects model (R nlme package) with random effects accounting for family. For the DLPFC dataset, we assumed individuals were unrelated and fit a simple linear model (R base package). In each model, we include cell-type proportion, age, age-squared, and sex as covariates. We introduced an additional covariate for diabetes status when regressing Matsuda index due to a known significant association between these two variables. We test whether the cell proportion-effect estimates deviate significantly from 0 using a $t$-test. Each R method implements the described model fitting and significance testing.

**Bisque marker-based decomposition model**. In order to estimate cell-type proportions across individuals without the use of a cell-type-specific gene

expression panel as reference, we use a weighted PCA approach. BisqueMarker requires a set of marker genes for each cell type as well as the specificity of each marker determined by the fold-change from a differential expression analysis. Typical single-cell RNA-seq workflows calculate marker genes and provide both $p$-values and fold-changes, as in Seurat[36]. For each cell type, we take statistically significant marker genes (FDR < 0.05) ranked by $p$-value. A weighted PCA is calculated on the expression matrix using a subset of the marker genes by first scaling the expression matrix and multiplying each gene column by its weight (the log-fold change) $\mathbf{XW}$, where $\mathbf{X}$ is the sample by gene expression matrix and $\mathbf{W}$ is a diagonal matrix with entries equal to log-fold change of the corresponding gene. The bulk expression $\mathbf{X}$ should be corrected for global covariates so that the proportion estimates do not reflect this global variation. The first PC calculated from $\mathbf{XW}$ is used as the estimate of the cell-type proportion. This allows cell-type-specific genes to be prioritized over more broadly expressed genes. Alternatively, if weights are not available, PCA can be run on the matrix $\mathbf{X}$ and the first PC can be used.

In order to select marker genes, we iteratively run the above PCA procedure on a specified range of markers (from 25 to 200) and calculate the ratio of the first eigenvalue to the second. We then select the number of marker genes to use that maximizes this ratio. This procedure is similar to other methods which select the number of markers to use by maximizing the condition number of the reference matrix[13].

**Software used**. Single-nucleus RNA-seq data were aligned using CellRanger 3.0.2 against the GRCh38.p12 genome assembly. Bulk RNA-seq data were aligned with STAR 2.5.1 and quantified using featureCounts 1.6.3, both against the GRCh38.p12 genome assembly. R 3.5.1 was used for further processing and decomposition experiments. The Seurat v3.0.0 R package was used to filter, cluster, and identify cell-type marker genes from the single-nucleus data. The sctransform 0.2.0 R package was used to normalize and scale the single-nucleus data. Bisque 1.0, xbioc 0.1.7, Biobase 2.4.2, MuSiC 0.1.1, bseqsc 1.0, CIBERSORT v1.06, and CIBERT-SORTx were all used for decomposition using the processed bulk and single-nucleus RNA-seq data. The R nlme 3.1-127 package was used for linear-mixed-model association. All visualizations and were generated with Python 3.7.2 using Seaborn 0.9.0, Matplotlib 3.0.3, Pandas 0.24.2, and Numpy 1.16.2, sklearn 0.20.3, and scipy 1.2.1.

**Reporting summary**. Further information on research design is available in the Nature Research Reporting Summary linked to this article.

## Data availability

The adipose data used in these analyses are available from the corresponding authors upon reasonable request. The DLPFC data are available on Synapse (10.7303/ syn3219045). Single-nucleus RNA-seq data (https://www.synapse.org/#!Synapse: syn16780177), bulk RNA-seq data (https://www.synapse.org/#!Synapse:syn3388564), and phenotypes (https://www.synapse.org/#!Synapse:syn3191087) are available under controlled use conditions set by human privacy regulations. A data use agreement is required to access these data. The source data underlying Tables 2 and 3, Figs. 2–5, Supplementary Tables 1 and 2, and Supplementary Figs. 1, 2, 3, 4, 5 are provided as a Source Data file.

## Code availability

Bisque is available as an R package named "BisqueRNA" that is available on CRAN and Bioconda. The source code for this package is available at https://github.com/cozygene/ bisque and is under the GPL-3 license.

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

## Acknowledgements

We thank the individuals who participated in the Finnish twin study, as well as Jaakko Kaprio and Aila Rissanen for the contributions to the twin study. We also thank the UNGC sequencing core at UCLA for performing RNA sequencing. This study was funded by National Institutes of Health (NIH) Grants HL-095056, HL-28481, and U01 DK105561. K.H.P. was supported by the Academy of Finland (272376, 266286, 314383, 315035), Finnish Medical Foundation, Finnish Diabetes Research Foundation, Novo Nordisk Foundation, Gyllenberg Foundation, Sigrid Juselius Foundation, Helsinki University Hospital Research Funds, and University of Helsinki. The funders had no role in study design, data collection, and analysis, decision to publish, or preparation of the article. The results published here are in whole or in part based on data obtained from the AMP-AD Knowledge Portal (https://doi.org/10.7303/syn2580853). Study data were provided by the Rush Alzheimer's Disease Center, Rush University Medical Center, Chicago. Data collection was supported through funding by NIA Grants P30AG10161, R01AG15819, R01AG17917, R01AG30146, R01AG36836, U01AG32984, U01AG46152, U01AG61356, the Illinois Department of Public Health, and the Translational Genomics Research Institute. B.J. was supported by the National Science Foundation Graduate Research Fellowship Program under Grant No. DGE-1650604. M.A. was supported by the HHMI Gilliam fellowship. Z.M. was supported by the AHA Grant 19PRE34430112. K.M.G. was supported by the NIH/NHLBI F31 fellowship HL142180. E.H., B.J., and E.R. were partially supported by the National Science Foundation (Grant No. 1705197). E.H., E.R., and B.J. were partially supported by NIH/NHGRI HG010505-02.

## Author contributions

E.H. and P.P. conceived of and supervised the study. B.J. and M.A. developed the method and associated software as well as carried out experiments. K.H.P. provided the subcutaneous adipose samples. M.A. and K.M.G. developed and performed the single-nucleus RNA-seq in the frozen subcutaneous adipose samples. J.H.S. provided guidance on the analysis of the dorsolateral prefrontal cortex dataset. E.R., Z.M., and A.K. provided statistical and biological insight for the method validation. B.J., M.A., P.P., and E.H. wrote the manuscript with support from all authors.

## Competing interests

The authors declare no competing interests.
