## [Peer Review File · Nature Communications]

Reviewers' comments:

Reviewer #1 (Remarks to the Author):

Jew et al. presented Bisque, a computational tool for NNLS (non-negative least squares) regression-based deconvolution of bulk expression data. Bisque uses transformation framework to enhance the gene-specific proportional relationship between reference scRNA-seq driven cell type specific reference profile and bulk RNA-seq measurements, that may be biased due to confounding effects such as technical variation in sample and library preparation and different single cell technology platforms. The transformation aims to match the mean and variance expression of the observed bulk across all patients to the mean and variance of the pseudo bulk expression that is calculated from the scRNA-seq.

Major Comments:

1. In single cell sequencing technology, there are difficulties to detect some specific cell types such as neutrophils, as previously reported (Smillie et al. 2018; Schelker et al. 2017), however these cell types may contribute to the measured bulk expression. This will likely present a problem in the comparison of the pseudo-bulk obtained from scRNA-seq to the bulk expression measurements. I don't expect this will be easy to overcome, but minimally I would say a cautionary note in the text is needed and ideally there may be a way to detect when this is a serious concern in which the algorithm could not be used (akin to an modelling an unknown cell-type but issuing some flag).
2. It would be of value to show using subsampling of real data or simulations how the estimated proportion prediction might be affected from the reference scRNA-seq sample size and variance.
3. reference marker profile:
 - a. How does the number of chosen markers in the selected cell specific reference profile effect proportion estimates performance?
 - b. How does the algorithm perform in estimating sub-populations with relatively high overlap between marker genes ? How does performance change as a function of marker overlapping compared to the other methods?
5. The authors show that the mean of cell proportions for the tested cohorts were similar to that of the scRNA-seq, suggesting good performance, but since the algorithm uses transformation that aim to match the mean of the bulk and scRNA-seq based pseudo-bulk, the cell proportion estimates performance should preferentially be tested using independent technology. The presented correlations between phenotype and proportion estimates, although significant, were relatively weak.

Minor comments:

Figure 3 and 4- Spearman's rho value should be presented with significance (as noted in the text).

Reviewer #2 (Remarks to the Author):

Jew et al. present a new tool "Bisque" for estimating cell type proportion in bulk RNA-seq data based on reference single-cell/single-nuclide expression profiles. Their method outperforms several existing approaches on simulated datasets and recovers some known biological features in real data.

Their manuscript is clear, and the method is interesting. However, I have a few concerns with the assumptions as well as general applicability of the tools. I have provided more details in my comments below.

1. My first concern is about the assumption made in the abstract. "Bisque requires a single-cell

reference data that reflects physiological cell type composition and can further leverage datasets that includes both bulk and single cell measurements over the same samples for improved accuracy." The assumption means that the cell type proportion from single cell data reflects the true cell type proportions in the bulk data. However, this assumption is usually violated, and this is why we want to perform deconvolution to estimate cell type proportions from bulk tissue rather than use the cell type proportions from single cell. Single-cell technology process introduced cell loss and the cell loss rate is usually different for each cell type. This is the reason why many existing deconvolution methods used the gene expression as reference matrix for deconvolution. For example, in Figure 2a, the plot on the left shows the discrepancies between single-cell based reference and bulk expression, the difference between bulk and artificial bulk might come from the cell type proportion shifts rather than technique bias.

Since Bisque still performs better than MuSiC, CIBERSORT and Bseq-sc on real data in this paper, I think that Bisque deconvolution results are probably not sensitive to the assumption of the cell types or the Bisque results are biased towards single cell proportions but still can recover the biological signals. I would further suggest that the authors perform a series of deconvolution using single-cell reference with different cell type proportions and see if Bisque still have better performance than other methods with different proportions from single cell reference.

2. Compared with other methods, Bisque has better performance in simulation settings as well as in real data. CIBERSORTx, an updated version of CIBERSORT with batch correction, gives comparable results as Bisque while CIBERSORT does not perform as good as CIBERSORTx. It seems to me that Bisque also can correct batch effect in deconvolution. Does Bisque have any strategies to address batch effects?

3. On page 11, the authors claimed that after merging subtypes, each method was able to quantify major cell population. Can you determine which group of genes that boost estimation accuracy of each subtype via Bisque transformation?

Reviewer #1 (Remarks to the Author):

Jew et al. presented Bisque, a computational tool for NNLS (non-negative least squares) regression-based deconvolution of bulk expression data. Bisque uses transformation framework to enhance the gene-specific proportional relationship between reference scRNA-seq driven cell type specific reference profile and bulk RNA-seq measurements, that may be biased due to confounding effects such as technical variation in sample and library preparation and different single cell technology platforms. The transformation aims to match the mean and variance expression of the observed bulk across all patients to the mean and variance of the pseudo bulk expression that is calculated from the scRNA-seq.

Major Comments:

1. In single cell sequencing technology, there are difficulties to detect some specific cell types such as neutrophils, as previously reported (Smillie et al. 2018; Schelker et al. 2017), however these cell types may contribute to the measured bulk expression. This will likely present a problem in the comparison of the pseudo-bulk obtained from scRNA-seq to the bulk expression measurements. I don't expect this will be easy to overcome, but minimally I would say a cautionary note in the text is needed and ideally there may be a way to detect when this is a serious concern in which the algorithm could not be used (akin to an modelling an unknown cell-type but issuing some flag).

We thank the Reviewer for this insightful comment and agree that discordance between scRNA-seq derived pseudo-bulk and bulk expression can arise from cell-type specific biases. We acknowledge that Bisque should ideally be used in scenarios where the single-cell data is sufficiently similar to the measured bulk expression (such as from the same tissue samples or without any known severe cell type biases). To address the possibility of an unknown cell population in the bulk expression data, we added the following simulation results to the “Robustness of the reference-based decomposition model” subsection of the manuscript results (page 14-15):

“Next, we simulated a situation where an unknown cell population contributes to bulk expression but is not represented in the snRNA-seq reference data. For situations where this unknown contribution varies across the bulk dataset, we simulated bulk expression by mixing the observed bulk expression for the DLPCF dataset with increasing amounts of expression observed in the adipose dataset. To determine the effect of unknown cell populations on our model, we analyzed the distribution of residual norms produced by the method. These residual norms provide a measure of the difference between the vector of observed bulk and expression reference weighted by the estimated proportions across all genes for each individual. As we increased the contribution from unknown cell types, the residual norm values tend to increase (Supplementary Fig. 5b). In our simulation

framework, this variability in unknown cell type contribution could be qualitatively identified by the presence of a multimodal residual norm distribution.”

2. It would be of value to show using subsampling of real data or simulations how the estimated proportion prediction might be affected from the reference scRNA-seq sample size and variance.

We agree with the Reviewer on the importance of the reference scRNA-seq sample size, since these datasets tend to be relatively small. We added the following results to the “Robustness of the reference-based decomposition model” subsection (page 15):

“Given that single-cell datasets still remain relatively small compared to bulk datasets, we also explored the impact of sample size in the reference single-cell data on the performance of Bisque. In the DLPFC dataset, we saw a drop in performance when using less than four randomly selected snRNA-seq samples (Supplementary Fig. 5c). This threshold is likely to differ between experiments, though we recommend at least three single-cell samples to generate reference data.”

3. reference marker profile:

a. How does the number of chosen markers in the selected cell specific reference profile effect proportion estimates performance?

Since marker gene selection can vary significantly between single-cell analysis pipelines, we agree that the effect of the number of markers should be explored. In the “Robustness of the reference-based decomposition model” subsection of our results, we added the following content to address marker gene selection (page 15-16):

“Finally, since marker gene selection can vary between studies, we were interested in the performance of Bisque as we varied the number of marker genes. Again, we measured cell type proportion estimation performance for microglia in the DLPFC dataset by correlating the estimates with Braak stage, which is known to have a positive association. We recalculated this correlation as we removed marker genes for this cell type. We removed marker genes in order of both decreasing and increasing log-fold-change, which provides a measure of the importance of marker genes for identifying this cell type. In both procedures, we observe that as we remove an increasing percentage of the 102 identified marker genes, performance remains stable until a shared drop off point around 75% (Supplementary Fig. 5d). Since we observed this trend in both marker gene removal schemes, we assume that a relatively few number of marker genes, regardless of their log-fold-change magnitude, can be used to accurately estimate cell type proportions. These results suggest that as long as a core set of marker

genes are present, variations in less important marker genes will have little effect on downstream analyses.”

b. How does the algorithm perform in estimating sub-populations with relatively high overlap between marker genes ? How does performance change as a function of marker overlapping compared to the other methods?

Compared to existing methods, Bisque was able to consistently infer the presence of sub-populations of major cell types in the cortex tissue. In this tissue, we estimated cell type proportions for 11 cell types that were derived from 6 major cell type populations. In this dataset, Bisque inferred the presence of each cell type whereas some existing methods provided estimates where only 1 subtype of each major cell populations were reported to be non-zero (Figure 4b). In the original submission, Supplementary Figure 4 compares the extent of marker gene overlap between sub-populations in the cortex tissue, which is greater than the overlap observed in the adipose tissue where existing methods could detect all desired cell types.

5. The authors show that the mean of cell proportions for the tested cohorts were similar to that of the scRNA-seq, suggesting good performance, but since the algorithm uses transformation that aim to match the mean of the bulk and scRNA-seq based pseudo-bulk, the cell proportion estimates performance should preferentially be tested using independent technology. The presented correlations between phenotype and proportion estimates, although significant, were relatively weak.

We recognize that the performance of cell type proportion estimates should preferentially be tested using an independent technology that does not have any biases. Unfortunately, the current technologies for determining cell type proportions especially in solid tissues, such as the adipose and brain tissues investigated here, each have their own biases and challenges. For example, flow cytometry of adipose tissue is technically very challenging when compared to applying flow cytometry to non-solid tissues, such as blood, and not all adipose cell types can be assessed using flow cytometry. Specifically, due to their high fat content, it is not possible to use flow cytometry to estimate the proportion of adipocytes which are the main cell type of adipose tissue. While histology and immunohistochemistry may also provide another independent validation, they are low-throughput and quantification of cell types is not straightforward when using these alternative techniques. We acknowledge that our method is designed to recover proportions reported by scRNA-seq or snRNA-seq technologies, the latter of which are known to be less biased than current scRNA-seq methods. To address these concerns, we edited the following excerpt from the discussion of the manuscript (page 19):

“In addition, a key assumption of our transformation framework is that single-cell based estimates of cell proportions accurately reflect the true proportions we wish to estimate. As a result of this assumption, Bisque provides estimates of cell

proportions reported by the single-cell technology used to generate the reference data. Given that snRNA-seq can provide less bias in isolating specific cell types compared to scRNA-seq (Wu et al. 2019, Bakken et al. 2018), we expect these estimates to be useful for downstream analyses such as those previously discussed. Nevertheless, the accuracy of Bisque may decrease if the proportion of cell types captured by single-cell experiments differs significantly from the true physiological distributions. Therefore, we advise users to take caution if there is a known significant bias in the single-cell measurements of a tissue, such as severe underrepresentation of a cell type of interest (Schelker et al. 2017, Bakken et al. 2018), that can affect downstream analysis. Our results demonstrate that even with these limitations, Bisque can be used to provide cell-type specific biological insight in relevant datasets.”

With respect to the strength of the correlations, while the general direction of correlation between the cell proportions and presented phenotypes are known, the strength of these correlations are not consistently reported. Therefore, we only use the direction and significance of the measured associations to measure performance.

Minor comments:

Figure 3 and 4- Spearman's rho value should be presented with significance (as noted in the text).

Reviewer #2 (Remarks to the Author):

Jew et al. present a new tool “Bisque” for estimating cell type proportion in bulk RNA-seq data based on reference single-cell/single-nuclide expression profiles. Their method outperforms several existing approaches on simulated datasets and recovers some known biological features in real data.

Their manuscript is clear, and the method is interesting. However, I have a few concerns with the assumptions as well as general applicability of the tools. I have provided more details in my comments below.

1. My first concern is about the assumption made in the abstract. “Bisque requires a single-cell reference data that reflects physiological cell type composition and can further leverage datasets that includes both bulk and single cell measurements over the same samples for improved accuracy.” The assumption means that the cell type proportion from single cell data reflects the true cell type proportions in the bulk data. However, this assumption is usually violated, and this is why we want to perform deconvolution to estimate cell type proportions from bulk tissue rather than use the cell type proportions from single cell. Single-cell technology process introduced cell loss and the cell loss rate is usually different for each cell type. This is the reason why many existing deconvolution methods used the gene expression as reference matrix for deconvolution. For example, in Figure 2a, the plot on the left shows the discrepancies

between single-cell based reference and bulk expression, the difference between bulk and artificial bulk might come from the cell type proportion shifts rather than technique bias. Since Bisque still performs better than MuSiC, CIBERSORT and Bseq-sc on real data in this paper, I think that Bisque deconvolution results are probably not sensitive to the assumption of the cell types or the Bisque results are biased towards single cell proportions but still can recover the biological signals. I would further suggest that the authors perform a series of deconvolution using single-cell reference with different cell type proportions and see if Bisque still has better performance than other methods with different proportions from single cell reference.

We thank the Reviewer for these constructive comments and agree that discrepancies between single-cell and bulk data may arise from biases in cell type capture. Therefore, we acknowledge to users that Bisque aims to recover cell type proportions measured by single-cell technologies, which may be biased but often maintain biological signals that may be of interest. We provide this information in the following revised discussion excerpt (page 19):

“In addition, a key assumption of our transformation framework is that single-cell based estimates of cell proportions accurately reflect the true proportions we wish to estimate. As a result of this assumption, Bisque provides estimates of cell proportions reported by the single-cell technology used to generate the reference data. Given that snRNA-seq can provide less bias in isolating specific cell types compared to scRNA-seq (Wu et al. 2019, Bakken et al. 2018), we expect these estimates to be useful for downstream analyses such as those previously discussed. Nevertheless, the accuracy of Bisque may decrease if the proportion of cell types captured by single-cell experiments differs significantly from the true physiological distributions. Therefore, we advise users to take caution if there is a known significant bias in the single-cell measurements of a tissue, such as severe underrepresentation of a cell type of interest (Schelker et al. 2017, Bakken et al. 2018), that can affect downstream analysis.”

Although absolute estimates of cell type proportions can be biased, we agree that we would like to see that the proportion-trait associations are still maintained. To explore the effects of cell type proportion biases, we added the following to the “Robustness of the reference-based decomposition model” subsection of our results (page 14):

“Our reference-based decomposition method is based on the assumption that cell populations are equally represented in single-cell and bulk RNA sequencing of the same tissue samples. Since this assumption may be violated (Schelker et al. 2017), we explored the performance of our model as we relaxed this assumption in simulations. First, we simulated snRNA-seq data where cell proportions were increasingly biased. Using the DLPFC snRNA-seq data, we downsampled or upsampled the cells identified as microglia at varying levels and performed

decomposition. Indeed, the absolute estimates produced by Bisque propagated these shifts in snRNA-seq proportions. However, we found that our estimated microglia proportions, regardless of these shifts, maintained an expected positive association with Braak stage. This positive association served as evidence for the correlation between these estimates and the true microglia proportions (Supplementary Fig. 5a). Given these results, we suggest that users take note of this behavior if both the mean abundances are important for downstream analysis and the single-cell reference data is known to be significantly biased against specific cell populations of interest.”

2. Compared with other methods, Bisque has better performance in simulation settings as well as in real data. CIBERSORTx, an updated version of CIBERSORT with batch correction, gives comparable results as Bisque while CIBERSORT does not perform as good as CIBERSORTx. It seems to me that Bisque also can correct batch effect in deconvolution. Does Bisque have any strategies to address batch effects?

We thank the reviewer for this comment. However, we would like to point out that CIBERSORTx actually does not correct for batch effects. Indeed, it uses ComBat as part of the algorithm, however the objective is not to correct for batches, but rather to address the biases. CIBERSORTx treats the single cell RNA as one batch and the bulk as another batch, and ComBat is then used to ‘merge’ or ‘correct’ these batches, thus accounting for the biases. Bisque addresses this bias through an efficient gene-specific linear transformation between the reference dataset and observed mixtures. This is related to batch correction, but it is not exactly the same objective.

3. On page 11, the authors claimed that after merging subtypes, each method was able to quantify major cell population. Can you determine which group of genes that boost estimation accuracy of each subtype via Bisque transformation?

The Bisque transformation procedure utilizes observed cell type proportions in single-cell data to calibrate its estimates. This calibration allows for detection of closely related cell types that can be distinguished in single-cell data in bulk expression. Furthermore, the magnitude of the Bisque transformations can provide insight into the group of genes that provide improved estimation accuracy. Specifically, genes with low correlation between single-cell and bulk experiments will have relatively small transformation coefficients and thus will have little effect on the non-negative least squares (NNLS) regression coefficients that are used as proportion estimates in Bisque. These genes that are discordant between the reference and bulk experiments may be used by existing methods and can decrease decomposition performance. As an illustration, we observe in the cortex datasets that most genes have a transformation coefficient near zero, indicating that these genes are not correlated between reference and bulk.

Reviewers' comments:

Reviewer #1 (Remarks to the Author):

Following the review, the authors added a subsection named 'Robustness of the reference-based decomposition model' and supplementary figure (Supplementary Figure 5) in which they tested and described by simulations the impact of sample size, the number of gene markers used and the effect of unknown cell population in the bulk expression on the decomposition performance.

Although the authors didn't test the decomposition performance using scRNA-seq independent technology, they tested the performance using rigorous leave-one out cross validation and they also added a cautionary note to the users in the text to carefully choose the reference scRNA-seq data, ensuring there is no significant cell population biases.

Reviewer #2 (Remarks to the Author):

I am happy to say that the authors have addressed most of my major concerns.

Comment 1: The simulations using biased cell type proportions is the best. Supplementary Figure 5a shows that the estimated proportions are almost coordinated with the reference proportions. Could you please make some comments or simulations on the linearity between reference and estimated proportions? Such as when will the estimated proportions become more stable with biased reference proportions?

I also have a minor comment: I think that Jew et al. can emphasize more on the novelty of using single-nuclei RNA-seq data as reference. It will benefit the readers more if the authors can make it clearer in the abstract.

Reviewer #1 (Remarks to the Author):

Following the review, the authors added a subsection named 'Robustness of the reference-based decomposition model' and supplementary figure (Supplementary Figure 5) in which they tested and described by simulations the impact of sample size, the number of gene markers used and the effect of unknown cell population in the bulk expression on the decomposition performance.

Although the authors didn't test the decomposition performance using scRNA-seq independent technology, they tested the performance using rigorous leave-one out cross validation and they also added a cautionary note to the users in the text to carefully choose the reference scRNA-seq data, ensuring there is no significant cell population biases.

We thank the reviewer for the helpful comments leading to the additional experiments that clarify the robustness of the method to readers.

Reviewer #2 (Remarks to the Author):

I am happy to say that the authors have addressed most of my major concerns.

Comment 1: The simulations using biased cell type proportions is the best. Supplementary Figure 5a shows that the estimated proportions are almost coordinated with the reference proportions. Could you please make some comments or simulations on the linearity between reference and estimated proportions? Such as when will the estimated proportions become more stable with biased reference proportions?

We thank the reviewer for the comments. The linearity between the reference and estimated proportion distributions stems from the assumption of the model that the reference dataset reflects the distribution that is observed in the bulk. In the results section of the manuscript describing the robustness of our method on page 14, we found that these simulated biases do not significantly affect cell type proportion correlations. However, we also advise users with the following in our previously revised manuscript:

“Given these results, we suggest that users take note of this behavior if both the mean abundances are important for downstream analysis and the single-cell reference data is known to be significantly biased against specific cell populations of interest.”

We would also like to point out that these biases that would affect this linearity have been found to be less significant in single-nucleus sequencing compared to single-cell, which is also noted in the original manuscript on page 19:

“Given that snRNA-seq can provide less bias in isolating specific cell types compared to scRNA-seq (Wu et al. 2019, Bakken et al. 2018), we expect these

estimates to be useful for downstream analyses such as those previously discussed.”

I also have a minor comment: I think that Jew et al. can emphasize more on the novelty of using single-nuclei RNA-seq data as reference. It will benefit the readers more if the authors can make it clearer in the abstract.

We agree with the reviewer and have updated the abstract with the following:

“Bisque implements a regression-based approach that utilizes single-cell RNA-seq (scRNA-seq) or single-nucleus RNA-seq (snRNA-seq) data to generate a reference expression profile and learn gene-specific bulk expression transformations to robustly decompose RNA-seq data.”

REVIEWERS' COMMENTS:

Reviewer #2 (Remarks to the Author):

The authors addressed my concerns and comments successfully. I have no further comments or questions.